# COVID-19 Stress and Food Intake: Protective and Risk Factors for Stress-Related Palatable Food Intake in U.S. Adults

**DOI:** 10.3390/nu13030901

**Published:** 2021-03-10

**Authors:** Jennifer R. Sadler, Gita Thapaliya, Elena Jansen, Anahys H. Aghababian, Kimberly R. Smith, Susan Carnell

**Affiliations:** 1Division of Child & Adolescent Psychiatry, Department of Psychiatry & Behavioral Sciences, Johns Hopkins University School of Medicine, Baltimore, MD 21205, USA; gthapal2@jhmi.edu (G.T.); elena.jansen@jhmi.edu (E.J.); aaghaba1@jhmi.edu (A.H.A.); susan.carnell@jhmi.edu (S.C.); 2Department of Psychiatry & Behavioral Sciences, Johns Hopkins University School of Medicine, Baltimore, MD 21205, USA; kimberly.smith@jhmi.edu

**Keywords:** COVID-19, food intake, stress, emotional overeating, cognitive flexibility

## Abstract

(1) Background: The coronavirus (COVID-19) pandemic has caused disruptions to what people eat, but the pandemic’s impact on diet varies between individuals. The goal of our study was to test whether pandemic-related stress was associated with food intake, and whether relationships between stress and intake were modified by appetitive and cognitive traits. (2) Methods: We cross-sectionally surveyed 428 adults to examine current intake frequency of various food types (sweets/desserts, savory snacks, fast food, fruits, and vegetables), changes to food intake during the pandemic, emotional overeating (EOE), cognitive flexibility (CF), and COVID-19-related stress. Models tested associations of stress, EOE, and CF with food intake frequency and changes to intake. (3) Results: Models demonstrated that the positive relationship between stress and intake of sweets/desserts was stronger with higher EOE, while the positive relationship between stress and intake of chips/savory snacks was weaker with higher CF. Higher EOE was associated with greater risk of increased intake of palatable foods. (4) Conclusions: Findings suggest that emotional overeating may escalate stress-associated intake of high-sugar foods, and cognitive flexibility may attenuate stress-associated intake of high-fat foods. Differences in appetitive and cognitive traits may explain changes to and variability in food intake during COVID-19, and efforts to decrease emotional overeating and encourage cognitive flexibility could help lessen the effect of COVID-19-related stress on energy dense food intake.

## 1. Introduction

The COVID-19 pandemic [1] has significantly increased stress and anxiety for many adults [2]. Factors such as health concerns, such as oneself or a loved-one contracting the virus; financial stress related to loss of income or the pandemic-induced economic downturn; disruptions to daily life, childcare, and education due to lockdown restrictions; and general uncertainty about the future and feelings of helplessness contribute to reports of increased stress [3,4]. However, while a majority of the global population has experienced pandemic-associated environmental stressors, there is substantial individual variation in the quantity and severity of stressors experienced, and in behavioral and physiological responses to those stressors [5,6]. Psychological characteristics such as cognitive flexibility may help reduce stress response [7]. Cognitive flexibility is defined by one’s ability to modify patterns of thought and behavior in response to a challenge, and high cognitive flexibility enables individuals to be resilient in the face of stress [8]. In the COVID-19 pandemic, greater psychological flexibility, or the ability to recognize and adapt to situational demands, has been related to lesser reported pandemic-related worry and distress in a sample of UK adults [9]. Greater cognitive or psychological flexibility may also help people to make adaptive behavioral responses to pandemic-associated stressors. Overall, initial research during COVID-19 suggests that individual characteristics such as cognitive flexibility may impact susceptibility to pandemic stress.

One potential behavioral response to environmental stress is to increase food intake [10,11], especially of palatable, energy dense foods [12,13,14]. Eating “comfort” foods in response to chronic stress can improve negative affect and decrease cortisol levels, especially among women and people reporting high stress proneness [15,16,17]. However, long-term adoption of this stress reduction strategy may be maladaptive as it could contribute to known associations of stress with obesity and related diseases [18,19]. Individuals may vary in their tendency to eat in response to stress. For example, adults with high emotional overeating, representing an individual’s tendency to overeat to suppress or soothe negative emotions, show greater food intake in response to acute stressors [20,21]. In contrast, for some individuals, acute stress acts to decrease intake [22,23]. Together, these findings suggest that both psychological (e.g., cognitive flexibility) and appetitive (e.g., emotional overeating) factors may make some individuals more likely to increase or decrease food intake due to pandemic-related acute and chronic stress.

Early evidence supports that the COVID-19 pandemic is associated with changes to self-reported food intake, with individuals reporting greater intake of sweets, desserts, salty snacks, and other palatable foods [24,25,26,27,28]. Women below the age of 29 years, and those with low cognitive restraint were more likely to report increased intake of high caloric foods [26,27], while positive coping strategies such as acceptance and cognitive reframing were protective against the pandemic-associated increases to palatable food intake [27]. Studies have also demonstrated positive associations of stress and increased food intake during the pandemic [25,26,27,28] but, to date, no study has examined whether individual characteristics may alter susceptibility to stress-induced changes to diet during the pandemic.

Understanding who is more or less vulnerable to obesogenic eating behaviors during the pandemic is critical to targeting public health interventions. Thus, in addition to describing intake of snack foods, fast food, and fruits and vegetables during the COVID-19 pandemic, we sought to identify how individual differences may exacerbate or protect against stress-related overeating during the COVID-19 pandemic. Specifically, we examined two characteristics previously associated with response to stress: emotional overeating and cognitive flexibility. We examined how COVID-19-related stress, emotional overeating, and cognitive flexibility as independent factors associated with food intake and changes to food intake during the pandemic. We also tested whether emotional overeating and cognitive flexibility interacted with COVID-19-related stress to influence food intake. We hypothesized that emotional overeating could increase an individual’s susceptibility to increasing intake in response to pandemic-associated stress, while cognitive flexibility may help people moderate tendencies to seek comfort in less healthy behaviors in service of a larger goal to maintain a healthy lifestyle through the pandemic. We therefore predicted that we would see a positive relationship between stress and intake of energy dense foods, and we hypothesized that this relationship would be stronger in those with higher emotional overeating, and weaker or absent in those with higher cognitive flexibility.

## 2. Materials and Methods

### 2.1. Survey Development and Sample Recruitment

An online survey was developed in the Spring of 2020 to examine the impact of COVID-19 on families’ eating behavior, stress, screen time, physical activity, and sleep. Data collection occurred in the months of May and June 2020. Recruitment occurred on social media and through Amazon Mechanical Turk (MTurk). The survey was targeted to adults at least 18 years of age. Participants recruited through MTurk received $6 compensation for survey completion, and participants recruited through social media were entered into a gift card lottery. Further information on survey development and recruitment can be found in [29]. All methods were approved by the Johns Hopkins University Institutional Review Board (protocol code 92328, approval date 26 May 2020).

### 2.2. Measures

#### 2.2.1. Demographics and Self-Reported Height and Weight

Participants provided demographic information including age, sex, employment status, education level, annual household income, food security, receipt of public assistance, and race/ethnicity. A socioeconomic disadvantage index [29] was created by summing four dichotomized indicators of relative disadvantage (i) lower household income (<$50,000 USD = 1, ≥$50,000 USD = 0), (ii) lower education (2 year college degree or less = 1, 4 year college or graduate degree = 0), (iii) food insecurity (yes = 1, no = 0), and (iv) receipt of public assistance (yes = 1, no = 0). The resulting score was a continuous variable (range 0–4) with higher scores reflecting greater disadvantage. Participants also self-reported height and weight, which were used to calculate body mass index (BMI).

#### 2.2.2. Self-Reported Current Food Intake and Changes to Intake

Food intake questions were adapted from food frequency questionnaires [30,31] to assess the frequency of consumption of specific food types. Frequency of intake in the past 7 days was assessed for sweets/desserts ((1a) chocolate or candies; (1b) cookies, cakes, pies, brownies; (1c) doughnuts, danishes, muffins; (1d) ice cream and frozen desserts), chips/savory snacks (including (2a) regular chips, (2b) low-fat chips, (2c) other salty snacks), and (3) food from fast food restaurants (e.g., McDonald’s, Burger King, Domino’s). Response options ranged from never to 6 or more times per day. Participants also reported the number of servings of fruits and vegetables consumed in the past 7 days. Standard serving sizes for a variety of fruits and vegetables were provided to increase accuracy with reporting. Response options ranged from 0 servings to 6 or more servings per day. All responses were recoded to reflect the frequency of consumption or number of servings consumed per week [32]. Participants also reported how intake of each food item in the past week compared with before the COVID-19 crisis. The response options ranged from 1 = “much less than before” to 5 = “much more than before”. Responses were collapsed into three categories: decreased intake (‘much less than before” and “a little less than before”), no change to intake (‘Same as before”), and increased intake (“a little more than before” and “much more than before”).

#### 2.2.3. COVID-19-Related Stress and Stress before COVID-19

Participants reported on stress related to the COVID-19 pandemic and general stress before the pandemic. To assess COVID-19-related stress, participants reported on 16 items assessing stress in relation to the pandemic (“How stressed are you about the following in relation to the COVID-19 crisis?”). Questions probed financial concerns (e.g., “Losing my job” and “Not being able to pay for basic needs”), concerns about health (e.g., “A relative (e.g., grandparent) or close family friend will get COVID-19” and “I will be unable to access medical care for myself or my family”), and concerns about life disruptions (e.g., “Ongoing need for social distancing” and “Decreased productivity at work”). Response options ranged from 1 = “not at all” to 5 = “extremely”. All items were averaged, and the overall mean COVID-19 stress score was used in analysis (Cronbach’s alpha = 0.91). To assess general stress before the pandemic, participants indicated how stressed they were in general before the crisis on a 0–10 scale, with higher scores indicating greater stress.

#### 2.2.4. Emotional Overeating

The emotional overeating subscale from the Adult Eating Behavior Questionnaire [33] was used to assess participants’ tendency to overeat when experiencing 5 negative emotions (annoyed, worries, upset, anxious, and angry). This 5-item subscale assesses agreement with statements such as “*I eat more when I’m annoyed*”, with response options ranging from 1 = “Strongly Disagree” to 5 = “Strongly Agree”. Emotional overeating was calculated as the mean response across the 5 items (Cronbach’s alpha = 0.92), where higher scores reflect higher emotional overeating.

#### 2.2.5. Cognitive Flexibility

The Cognitive Flexibility Scale [34] was used to assess participants’ self-efficacy in adapting to new information. This 12-item scale assesses agreement with statements such as “*My behavior is a result of conscious decisions that I make*”, “*I can find workable solutions to seemingly unsolvable problems*”, and “*I avoid new and unusual situations*” [reverse scored]), with response options on the scale ranging from 1 = “Strongly disagree” to 6 = “Strongly agree”. Responses on the 12 items were summed to generate a cognitive flexibility score that could range from 12 to 72 (Cronbach’s alpha = 0.87), with higher values reflecting greater cognitive flexibility.

### 2.3. Data Analysis

Descriptive statistics and bivariate correlations among variables were examined using RStudio (R version 4.0.3; The R Foundation for Statistical Computing).

Multivariate linear regression was used to model associations of COVID-19-related stress, emotional overeating, and cognitive flexibility, with intake frequency for each food category. Regression analyses used mean-centered variables to improve interpretability of estimates. All models controlled for the following covariates: age, sex, BMI, general stress before the pandemic, and socioeconomic disadvantage score. We implemented a data-driven, model-building procedure using forward selection, allowing us to test interactions between stress and emotional overeating, between stress and cognitive flexibility, and the three-way interaction of stress, emotional overeating, and cognitive flexibility. The base model for a given food intake frequency outcome included stress, emotional overeating, cognitive flexibility, and covariates as independent variables without interaction terms (Model 1). Pairwise interactions between stress and emotional overeating, and stress and cognitive flexibility, were then added to the model (Model 2). In the next step, a three way interaction between stress, emotional overeating, and cognitive flexibility was added (Model 3). With the addition of each interaction term, an analysis of variance (ANOVA) test was used to determine if the model was significantly better at capturing the data than the previous model. The threshold for significant model improvement was *p* < 0.05. A similar procedure was employed to test for associations of COVID-19-related stress, emotional overeating, and cognitive flexibility, with self-reported changes to intake frequency for each food type. Models were run separately for individual food items in the sweets/desserts and chips/savory snacks categories. Multinomial logistic regression was implemented in R’s ‘nnet’ package to test associations of COVID-19-related stress, emotional overeating, and cognitive flexibility with the relative risk of increasing food intake or decreasing food intake as compared to no change to intake. Akaike information criterion (AIC) value was used to compare between Models 1, 2, and 3, where the model with the lowest AIC was selected as the final model. Relative risk ratios and confidence intervals are reported. The threshold for model and independent variable significance was *p* < 0.05. To visualize significant interactions of stress with cognitive flexibility or emotional overeating, the mean and standard deviation of cognitive flexibility/emotional overeating was used to split the sample into low (<-1SD from mean), mid (-1SD to +1SD from mean), and high (>+1SD from mean) groups and the association between stress and intake frequency was plotted by group.

### 2.4. Exploratory Analysis of Possible COVID Cases

Two survey questions assessed whether participants had COVID since January 2020. The first asked “Has a healthcare provider ever told you that you had, or might have had, COVID-19” and the second asked “Have you ever tested positive for COVID-19?”. Finally, a third question asked “Are you currently experiencing a loss of sense of smell or taste”. To examine the impact of possible COVID-19, participants who responded affirmatively to any of these questions were classified as “possible/confirmed COVID-19 cases”. Demographic characteristics, appetitive and psychological traits, and food intake outcomes were compared between the possible/confirmed COVID-19 subsample and the rest of the sample. Additionally, possible/confirmed COVID-19 was added as a covariate in multivariate linear regression models and multinomial logistic regression models to determine the impact of controlling for possible/confirmed COVID-19 on the associations of COVID-19-related stress, emotional overeating, and cognitive flexibility and food intake/intake changes during the pandemic.

## 3. Results

### 3.1. Sample

A total of 579 individuals completed the survey. Of that sample, 428 participants provided complete data on COVID-19-related stress, cognitive flexibility, emotional overeating and covariates, representing the analytic sample. Demographic characteristics of the sample are shown in Table 1.

The average survey respondent was 37.5 years old, female, Non-Hispanic White, at an overweight BMI (mean 27.9 kg/m^2^) and endorsed one form of socioeconomic disadvantage (mean score 1.0). Correlations between participant characteristics including COVID-19-related stress, emotional overeating, and cognitive flexibility can be found in Table 2. COVID-19-related stress was positively correlated with emotional overeating (r = 0.53, *p* < 0.001) and weakly negatively correlated with cognitive flexibility (r = −0.11, *p* = 0.03). Emotional overeating and cognitive flexibility were negatively correlated (r = −0.30, *p* < 0.001). Unadjusted Pearson’s correlation coefficients showed that COVID-19-related stress was significantly positively correlated with sweets/dessert intake (r = 0.23, *p* < 0.001) and chips/savory snack intake (r = 0.20, *p* < 0.001). Emotional overeating was positively correlated with sweets/dessert intake (r = 0.31, *p* < 0.001), chips/savory snack intake (r = 0.26, *p* < 0.001), and fast food intake (r = 0.17, *p* < 0.001). Cognitive flexibility was negatively correlated with chips/savory snack intake (r = −0.17, *p* < 0.001).

### 3.2. Self-Reported Current Food Intake and Change in Intake of Sweets, Snacks, Fast Food, Fruits and Vegetables during COVID-19

A summary of food intake frequency of sweets/desserts, savory snacks, fast food, fruits, and vegetables during COVID-19 is shown in Table 3. Self-reported intake frequency was the highest for sweets/desserts (consumed 17.4 times per week), then chips/savory snacks (consumed 13.9 times per week), followed by vegetables (11.9 servings per week) and fruits (9.61 servings per week). Intake frequency of fast foods was the lowest (consumed 1.54 times per week) on average. Data on self-reported change in intake demonstrated that intake of sweets/desserts underwent the greatest increase, with 40.9% of the sample reporting increased intake during the pandemic (including “a little more than before” and “much more than before” responses). Conversely, intake of fast foods showed the greatest decrease, with 39.0% of the sample reporting decreased intake during the pandemic (including “a little less than before” and “much less than before” responses). Intake of fruits and vegetables were the least affected by the pandemic, with 52.1% and 56.5% of the sample reporting no change in intake compared with pre-pandemic consumption.

### 3.3. Association of COVID-19-Related Stress, Emotional Overeating and Cognitive Flexibility with Intake during COVID-19

The best-fitting models associated with food intake frequency during the pandemic are shown in Table 4.

Significant interactions between COVID-19-related stress, emotional overeating, and cognitive flexibility were found across all food types except fruit and vegetable intake. There was a significant positive interaction between COVID-19-related stress and emotional overeating for intake frequency of sweets/desserts, such that individuals with greater emotional overeating showed a stronger positive relationship between COVID-19-related stress and sweets/desserts intake as compared with those with low emotional overeating (Figure 1A). Significant interactions between COVID-19-related stress and cognitive flexibility were associated with intake frequency of chips/savory snacks and fast foods. The interaction for chips/savory snacks was negative, meaning greater cognitive flexibility was associated with a weaker positive relationship between COVID-19-related stress and chips/savory snack intake (Figure 1B). A similar effect was observed for fast foods; the negative interaction term reflected that greater cognitive flexibility was associated with a weaker positive relationship between COVID-19-related stress and fast food intake. Fruit intake was not associated with COVID-19-related stress, emotional overeating, or cognitive flexibility, while vegetable intake was negatively related to emotional overeating. We did not find evidence for significant three-way COVID-19-related stress x emotional overeating x cognitive flexibility interactions for intake of any food type. Full final models with covariates can be found in Appendix A.

Figure 1 Associations of COVID-19-related stress, emotional overeating, and cognitive flexibility with food intake (*n* = 428).

### 3.4. Association of COVID-19-Related Stress, Emotional Overeating and Cognitive Flexibility with Changes to Food Intake during COVID-19

Multinomial logistic regression models (Table 5) showed evidence for associations between emotional overeating and increased intake of sweets and desserts and chips and savory snacks.

With the ‘no change’ group as reference, increasing emotional overeating was significantly associated with the relative risk of increasing intake of chocolates and candies (risk ratio [RR] = 1.64; 95% confidence interval [CI] = [1.22–2.2]); cookies, cakes, pies, and brownies (RR = 1.51; CI = [1.13–2.03]); donuts, danishes, and muffins (RR = 1.56; CI = [1.13–2.15]); and ice cream and frozen desserts (RR = 1.73; CI = [1.29–2.32]). The relative risk ratio means that for a one unit increase in emotional overeating, the risk of reporting increase intake of chocolates and candies was 1.64 times higher than the risk of reporting no change to intake. Similar associations were observed for savory palatable foods, where increasing emotional overeating was significantly associated with the relative risk of increasing intake of regular chips (risk ratio [RR] = 1.64; 95% confidence interval [CI] = [1.22–2.2]); savory snack foods (RR = 1.51; CI = [1.13–2.03]); and fast foods (RR = 1.74; CI = [1.25–2.43]).

COVID-19-related stress also showed significant effects on the risk of increasing or decreasing intake during the pandemic for multiple food types. Increasing COVID-19-related stress was associated with an increased risk for both increased and decreased intake of cookies, cakes, pies, and brownies (increased intake: RR = 1.62; CI = [1.17–2.26]; decreased intake: RR = 1.48; CI = [1.06–2.07]) and ice cream and frozen desserts (increased intake: RR = 1.39; CI = [1–1.92]; decreased intake: RR = 1.42; CI = [1.01–1.99]). COVID-19-related stress was also positively associated with the risk of increased intake of low-fat chips (RR = 1.71; CI = [1.1–2.65]) and negatively associated with the risk of decreased intake of regular chips (RR = 1.45; CI = [1.01–2.08]), savory snacks (RR = 1.53; CI = [1.07–2.19]), fruits (RR = 1.83; CI = [1.29–2.58]), and vegetables (RR = 1.48; CI = [1.04–2.08]). Full final multinomial logistic regression with covariates can be found in Appendix B.

### 3.5. Exploratory Analysis of Possible/Confirmed COVID-19 Cases

Thirty (6.8% of total sample) participants reported possible or confirmed COVID-19. Of this subsample, 24 (80%) endorsed that a medical provider told them that they had or might have had COVID-19, 4 (13.3%) received a positive COVID-19 test, and 13 (43.3%) were currently experiencing a loss of smell or taste. There were no significant differences in the possible/confirmed COVID-19 subgroup and the rest of the sample in emotional overeating (*p* = 0.25), cognitive flexibility (*p* = 0.06), age (*p* = 0.65), sex (*p* = 0.82), BMI (*p* = 0.30), or pre-COVID stress (*p* = 0.09). However, the possible/confirmed COVID-19 subgroup had significantly higher COVID-19-related stress scores than the main sample (subgroup mean = 3.1; main sample mean = 2.6; t = −2.5; df = 32.3; *p* = 0.016). The possible/confirmed COVID-19 subgroup reported significantly more frequent intake of chips/savory snacks compared to the main sample (subgroup mean = 27.8; main sample mean = 12.9; t = −2.4; df = 30.0; *p* = 0.021). There were no differences in intake frequency for all other food types between the COVID-19 subgroup and the rest of the sample, and the two groups did not differ in reported changes to food intake during the pandemic.

When we controlled for possible/confirmed COVID-19 in our multivariate linear regression models of food intake frequency during the pandemic, the models of sweets/dessert intake, fast food intake, fruit intake, and vegetable intake were unchanged. Conversely, the model of chip/savory snack intake did change when we controlled for possible/confirmed COVID-19. Specifically, the interaction term for COVID-19-related stress and cognitive flexibility showed a weaker effect, where the interaction was still negative (estimate = −0.220), but trended towards statistical significance (*p* = 0.053). Multinomial logistic regression models of changes to food intake during the pandemic showed no differences in effects when possible/confirmed COVID-19 was added to the model as a covariate.

## 4. Discussion

Our study aimed to describe intake of snack foods, fast food, and fruits and vegetables during the COVID-19 pandemic, and identify cognitive and appetitive traits that exacerbate or protect against stress-related overeating during the pandemic in a sample of U.S. adults. Compared with fast foods and fruits and vegetables, intake frequency of snack foods and desserts was high during the pandemic. However, self-reported changes in snack food intake varied widely across the sample, with some participants reporting increases in snack food intake and others reporting decreases. When we examined factors associated with intake during the pandemic, we found that individual differences in COVID-19-related stress, emotional overeating, and cognitive flexibility were associated with intake frequency, such that emotional overeating strengthened the association between stress and intake frequency of sweets and desserts, while cognitive flexibility weakened the association between stress and savory snack intake. Emotional overeating was also associated with the higher relative risk of increasing the frequency of intake of sweets and desserts, chips and savory snacks, and fast food during the pandemic. Together, our results support that the COVID-19 pandemic had disparate effects on intake of snack foods, fast foods, and fruits and vegetables, and that interactions of COVID-19-related stress with appetitive and cognitive traits may explain individual differences in sweets and snack intake during the pandemic.

In general, intake of energy dense, processed snack foods was high during the COVID-19 pandemic in our sample. Survey respondents reported frequent intake of sweets and desserts and chips and savory snacks, with an average consumption of 17 times per week and 14 times in the previous week respectively. This equates to consuming sweets and desserts almost 2.5 times per day and chips and savory snacks 2 times per day. Frequent intake of sweets and desserts likely exceeds dietary recommendations to consume fewer than 10% of daily caloric intake from added sugars, and frequent intake of chips and savory snacks contributes to higher sodium and fat intake [35]. These results add to increasing evidence of high snack and sweets intake during the pandemic [28,36,37]. It is possible that increased intake of snack foods may be related to increased purchasing of snack foods that are shelf-stable and in large supply [38]. Further, frequent consumption of energy dense snack foods has been suggested to reflect the drive to soothe negative emotions such as stress and boredom associated with the pandemic and associated lockdown measures [27]. One study found that in a sample of Italian adults, approximately 50% of respondents reported “using food to respond to anxious feelings” and 55% reported a “need to increase food intake to feel better” [39], suggesting that for some individuals, stress during the pandemic increased food intake, especially of palatable foods. In our sample, unadjusted correlations revealed that COVID-19-related stress was positively associated with intake of sweets and desserts and chips and savory snacks. While the correlations were small, they suggest that among survey respondents, increasing stress specific to the COVID-19 pandemic was associated with increasing snack food intake. In support of this, COVID-19-related stress was also associated positively with the risk of reporting increased intake of palatable foods such as chocolate and candies and cookies, cakes, pies, and brownies during the pandemic. However stress was not consistently associated with increased intake during the pandemic. Increasing COVID-19-related stress was related to a higher risk of decreased intake of some palatable foods (regular chips and savory snacks) and fruits and vegetables. In both clinical and preclinical research, stress shows anorexigenic effects [10,40,41], and in humans, the negative effect of stress on intake appears to be more closely related to healthy foods such as fruits and vegetables [10,42]. Our results echo that of prior research [10,27,39,40,41], suggesting that the pandemic increased perceived stress and presented challenges to healthy eating behavior by increasing the drive to consume “comfort” foods and decreasing intake of fruits and vegetables.

To further understand how stress may lead some individuals to increase snack food intake, our study is the first to test how COVID-19-related stress interacts with emotional overeating and is consequently associated with snack intake during the pandemic. We found that intake of sweets and desserts was associated with a positive interaction between stress and emotional overeating such that for individuals with high emotional overeating, the positive relationship between COVID-19-related stress and sweets/dessert intake was stronger. Our model shows that being one standard deviation above the mean for COVID-19-related stress and the mean for emotional overeating is associated with an estimated 10 additional instances of sweets/dessert intake per week. Assuming a standard portion is consumed, this translates to consumption of approximately 250 additional kilocalories per day, which for the average participant in our study would result in a weight gain of approximately 12 kg over a year [43]. When we examined the relationship between COVID-19-related stress, emotional overeating, and cognitive flexibility, we found consistent evidence for an effect of emotional overeating on changes to palatable food intake frequency during the pandemic, where higher emotional overeating was related to a greater likelihood of increased intake of sweets and desserts, regular chips, savory snacks, and fast foods. Overall, our results suggest that people with high emotional overeating may be more susceptible to stress-induced overeating of sweets and desserts that puts them at a risk of weight gain. A similar effect of emotional overeating on stress and food intake has been previously observed in a laboratory setting, where emotional overeating was associated with greater intake of high-fat snacks in response to stress [23]. Of note, the interaction of emotional overeating and COVID-19-related stress was not a significant predictor of chips and savory snack intake, so in the present data, the effect of emotional overeating and stress on intake is specific to high fat, high sugar sweet food. This is consistent with previous research suggesting that stress specifically potentiates preferences for sweet foods [20,22]. When stressed, emotional eaters are more likely to consume palatable, energy dense foods, such as sweets and desserts, than unstressed and non-emotional eaters [20]. Further, stress is often implicated as a trigger for binge eating [44] and stressful life events often precede the development of binge eating disorder [45], which is characterized by binges on high energy dense foods, often sweet foods [46]. It is possible that emotional overeating may have increased in some individuals in response to stressors during the pandemic as a coping mechanism. Further research, especially studies with longitudinal measures are needed to understand how stress and emotional overeating continuously influenced intake during the pandemic.

Results also revealed that cognitive flexibility interacted with COVID-19-related stress to influence intake of chips and savory snacks and fast foods during the pandemic. Cognitive flexibility attenuated the association between COVID-19-related stress and intake of chips and savory snacks and fast foods such that those with high cognitive flexibility showed a weaker positive relationship between stress and chip/savory snack intake and fast food intake. Cognitive flexibility is a domain within executive function [47] and stronger executive functioning is thought to improve self-regulation of food intake [48]. Further, high fat food, such as fast food and chips/savory snacks, may decrease executive functioning performance [49,50]. While the role of executive function in diet is well studied, the relation of cognitive flexibility to intake is less established. One study reported a positive association of cognitive flexibility and fruit and vegetable intake in adults [51], while another found that cognitive flexibility was related to better adherence to dietary goals [52]. Notably, a number of studies show that stress impairs cognitive flexibility [53,54,55,56]. Our sample showed a weak, non-significant negative association between cognitive flexibility and COVID-19-related stress. Our finding suggests that for those with high cognitive flexibility it may be easier to counteract COVID-19-related stress by engaging executive functioning to support lower high fat food intake during the pandemic.

The effects observed in our analyses were largely not related to BMI. In multivariate regression models of food intake frequency during the pandemic, BMI was associated with less frequent fruit intake, but showed no association with the frequency of intake of other food types. Other research shows mixed associations between BMI and food intake during the pandemic. One study identified that BMI was positively associated with increased intake of palatable foods including packaged sweets, baked products, sweet beverages, savory snacks, and dressing and sauces during the pandemic [57], while another found no effect of BMI on changes to intake of various food groups [27]. In our sample, BMI showed no independent associations with intake of the food types examined, suggesting that in the present sample, BMI was not related to food intake frequency during the pandemic.

We observed wide individual variation in how the COVID-19 pandemic impacted food intake. Over one-third of the sample reported increased intake of sweets and desserts as compared with before the pandemic, but another third of the sample reported decreased intake of sweets and desserts in the same timeframe. Other research reports similar variability in COVID-19-related changes to diet, with some individuals reporting healthy dietary changes and other reporting shifts to unhealthy patterns of intake [24,27,58,59]. For example, one study found that individuals with higher health and diet literacy were more likely to increase healthy eating during the pandemic [60], while another reported that pro-healthy changes to diet were associated with older age and increased consumption of home cooking [58]. Another key element in dietary changes during the pandemic appears to be the role of snacking between meals [61,62,63]. While our survey did not assess whether snacking increased during the COVID-19 pandemic, the high proportion of participants reporting increased intake of snack foods and desserts suggests that intake outside of regular mealtimes could have increased in this sample as well.

Of note, the results here show that fruit and vegetable intake was relatively unchanged during the COVID-19 pandemic, with over half of the sample reporting no change to intake as compared with before the pandemic. Previous findings report increased intake of fruits and vegetables during COVID-19 lockdown [27], while others show decreased intake during COVID-19 lockdown [24,63]. At least one other study reported no changes to fruit and vegetable intake in the majority of their sample [58]. Differences may be related to differences in survey samples, specifically participant’s nationality and the lockdown measures in place when data were collected. The present sample included American adults and data were collected during a stage of the pandemic (May–June 2020) when lockdown measures varied widely across US states. While fruit and vegetable supply chains are especially susceptible to transport and logistic challenges during the pandemic [64], disruptions to food supply chains were minimal during this period [65], so access to fruits and vegetables was likely unchanged for the present sample during the period of data collection. Further, fruit and vegetable intake was not associated with COVID-19-related stress, suggesting that intake of fruits and vegetables was less affected by the COVID-19 pandemic than other types of food.

Initial evidence demonstrates that coronavirus infection and COVID-19 diagnosis can impact diet. Symptoms of COVID-19 such as anosmia and the treatment of COVID-19 increases the risk of dysphagia and malnutrition due to nausea, diarrhea, and the loss of appetite [66]. In an exploratory analysis, we identified thirty participants who had a confirmed case of COVID-19 or possibly had COVID-19 (either as indicated by a medical provider or by current loss of taste and smell). We found that unsurprisingly, for those who had or may have contracted COVID-19, stress related to the pandemic was significantly higher compared to the rest of our sample. In general, food intake frequency and changes to food intake during the pandemic did not differ between the possible/confirmed COVID-19 subgroup and the main sample. However, the possible/confirmed COVID-19 subgroup reported significantly more frequent intake of chips and savory snacks. It is unclear why the COVID-19 sample reported higher intake of chips and savory snacks. COVID-19 diagnosis and the duration of illness are associated with weight loss [67], but to date, little is known regarding how COVID-19 impacts diet during illness and during recovery. Further research in this area, especially within COVID-19 survivors is key for understanding the short and long term effects of COVID-19 on food intake.

The use of self-reported intake frequency of sweets, snack foods, and fast food is susceptible to errors in reporting [68,69]. In the present survey, participants reported the frequency of intake of specific foods, without reporting any information about the amount of food consumed. Self-served portion sizes can vary significantly between individuals [70], so it is unknown how many calories an individual consumed on any given instance of intake. For fruit and vegetable intake, this limitation was addressed by measuring the number of servings consumed in a week, and by providing participants with example serving sizes for a range of fruits and vegetables. Further research using more robust measures of intake (e.g., more detailed food frequency questionnaire or multiple 24 h food recalls [71] are recommended to improve understanding of how the COVID-19 pandemic has changed food intake. Similarly, participants completed a self-reported measure of how intake of food types during the pandemic compared with intake before the pandemic, rather than retrospectively reporting a measure of intake frequency before the pandemic. Because the delineation between responses is subjective (e.g., people may vary in what they consider a little more versus much more), bias could confound results. Most importantly, the data presented here are cross-sectional, limiting our ability to make inferences about the direction of effects of stress, cognition, and appetitive traits on diet. While most research on diet during the COVID-19 pandemic are cross-sectional, the opportunity to collect longitudinal data is expanding as the pandemic continues. Given the length of the pandemic, it is critical for researchers to collect measurements over time to better understand how cognitive flexibility may change with sustained chronic stress related to the pandemic, and whether appetitive traits change over the course of the pandemic or whether their effects as moderators of stress-related intake strengthen or diminish over time.

The present results highlight the importance of positive adaptive strategies to weather stress, boredom, and other negative emotions brought on by the pandemic. In our sample, cognitive flexibility is associated with better ability to control stress-associated intake of high fat fast foods and snack foods. The positive effects of cognitive flexibility on wellness during the pandemic extends beyond diet; in a sample of Italian adults, cognitive flexibility mitigated the detrimental impacts of COVID-19 risk factors (e.g., preexisting medical conditions) on mental health during lockdown [72]. Independent of the pandemic, cognitive training has beneficial effects on diet by promoting healthy food choices [73]. Thus cognitive training targeted at improving cognitive flexibility, such as acceptance and commitment therapy [74], may help attenuate the negative impact of the COVID-19 pandemic on diet in adults. Emotion regulation interventions targeted at preventing emotional overeating [75] and interventions that encourage positive emotional coping strategies [76] could also help those with high emotional overeating avoid excessive intake of palatable foods during the pandemic. In conclusion, our findings highlight the variability in individual dietary changes during the pandemic and demonstrate that cognitive and appetitive characteristics can impact the relationship between stress during the pandemic and intake of highly processed, energy dense foods.

## Figures and Tables

**Figure 1 nutrients-13-00901-f001:**
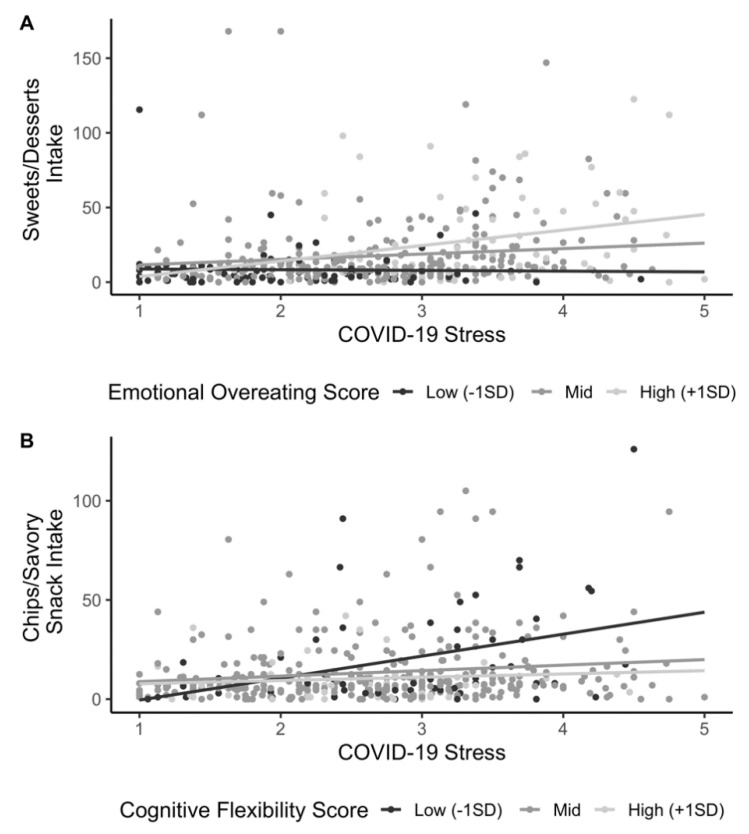
(**A**) Visualization of the significant interaction of emotional overeating and COVID-19-related stress on sweets/desserts intake. Lines depict the relationship between COVID-19-related stress and sweets/desserts intake across low (<-1SD from mean), mid (-1SD to +1SD from mean), and high (>+1 SD from mean) levels of emotional overeating. (**B**) Visualization of the significant interaction of cognitive flexibility and COVID-19-related stress on chips/savory snacks intake. Lines depict the relationship between COVID-19-related stress and chips/savory snacks intake across low (<-1SD from mean), mid (−1 to +1 SD from mean), and high (>+1 SD from mean) levels of cognitive flexibility.

**Table 1 nutrients-13-00901-t001:** Participant demographic characteristics (*n* = 428).

Characteristic	Min–Max	Mean (SD)
Age (years)	18–67	37.5 (8.25)
BMI (kg/m^2^)	15.8–58.9	27.9 (6.62)
COVID-19-Related Stress	1–5	2.7 (0.88)
Emotional Overeating Score	1–5	2.3 (1.02)
Cognitive Flexibility Score	15–66	49.7 (7.85)
Socioeconomic Disadvantage Score	0–4	1.0 (1.12)
Pre-COVID General Stress	0–9.5	3.85 (2.27)
Characteristic	Count	Percent
Sex		
Male	158	36.9%
Female	270	63.1%
Race		
Black or African American	27	6.3%
Indian American or Native Alaskan	5	1.2%
Native Hawaiian or Pacific Islander	2	0.5%
Asian	37	8.6%
Hispanic or Latin	15	3.5%
White	319	74.5%
Other	3	0.7%
More than one race	20	4.7%
Ethnicity		
Hispanic	41	9.6%
Non-Hispanic	382	89.3%
I don’t know	3	0.7%
Prefer not to answer	2	0.5%

**Table 2 nutrients-13-00901-t002:** Pearson’s correlation of independent variables, covariates, and food intake outcomes (*n =* 428) ^1^.

	Variable	1	2	3	4	5	6	7	8	9	10	11	12
1	COVID-19 Stress		−0.11	**0.53**	**0.33**	**0.16**	0.02	**0.22**	**0.23**	**0.2**	0.14	−0.01	0
2	CFS ^2^	0.025		**−0.3**	−0.13	0.04	0.13	−0.16	−0.15	**−0.17**	−0.15	−0.03	0.14
3	EOE ^3^	0	0		**0.32**	**0.19**	0.02	**0.2**	**0.31**	**0.26**	**0.17**	−0.06	−0.13
4	Pre-COVID Stress	0	0.006	0		0.08	−0.04	0.06	**0.16**	**0.19**	0.08	0.08	0.01
5	BMI ^4^	0.001	0.431	0	0.082		**0.17**	0.13	0.06	0.01	0.09	−0.11	−0.05
6	Age	0.676	0.008	0.713	0.423	0		−0.14	−0.01	0.03	0.03	0.05	0.12
7	SES disadvantage ^5^	0	0.001	0	0.24	0.006	0.003		0.1	0.05	0.12	−0.04	−0.05
8	Sweets/Desserts Intake	0	0.002	0	0.001	0.204	0.898	0.044		**0.54**	**0.34**	**0.16**	−0.04
9	Chips/Savory Snack Intake	0	0	0	0	0.894	0.535	0.315	0		**0.38**	0.15	0.01
10	Fast Food Intake	0.005	0.002	0.001	0.08	0.066	0.575	0.016	0	0		0.07	−0.01
11	Fruit Intake	0.827	0.502	0.238	0.089	0.02	0.343	0.4	0.001	0.003	0.16		**0.39**
12	Vegetable Intake	0.939	0.005	0.007	0.802	0.295	0.017	0.267	0.469	0.764	0.826	0	

^1^ Above diagonal: correlation coefficient (r); bold = *p* < 0.001; below diagonal: *p*-value; ^2^ CFS = cognitive flexibility score; ^3^ EOE = emotional overeating; ^4^ BMI = Body Mass Index; ^5^ SES = Socioeconomic status.

**Table 3 nutrients-13-00901-t003:** Self-reported current food intake and change in food intake during COVID-19 (*n =* 428).

		Self-Reported Intake Change during COVID-19		
**Times Consumed per Week**	**Mean (SD)**	*Decreased*	*No Change*	*Increased*
Sweets/Desserts	17.4 (23.8)	149 (34.8%)	104 (24.3%)	175 (40.9%)
Chips/Savory snacks	13.9 (17.9)	98 (22.9%)	186 (43.5%)	144 (33.6%)
Fast foods	1.54 (2.61)	167 (39.0%)	163 (38.1%)	98 (22.9%)
**Portions Consumed per Week**	**Mean (SD)**			
Fruit	9.61 (8.57)	98 (22.9%)	223 (52.1%)	107 (25.0%)
Vegetables	11.9 (9.65)	89 (20.8%)	242 (56.5%)	96 (22.4%)

**Table 4 nutrients-13-00901-t004:** Best-fit multivariate linear regression models of COVID-19-related stress, emotional overeating, and cognitive flexibility associated with food intake frequency during COVID-19 (*n =* 428).

Outcome	Independent Variables	Estimate	Std. Error	t-Value	*p*-Value ^3^
Sweets/Desserts	COVID-19 Stress	2.35	1.50	1.56	0.119
	EOE ^1^	4.68	1.37	3.42	**0.001**
	CFS ^2^	−0.14	0.15	−0.95	0.343
	COVID-19 Stress x EOE	4.03	1.23	3.27	**0.001**
Chips/Savory snacks	COVID-19 Stress	1.63	1.13	1.44	0.150
	EOE	3.05	1.02	3.00	**0.003**
	CF	−0.22	0.11	−1.91	0.057
	COVID-19 Stress x CFS	−0.28	0.11	−2.46	**0.014**
Fast foods	COVID-19 Stress	3.01	0.85	3.52	**0.000**
	EOE	0.17	0.15	1.10	0.274
	CFS	0.11	0.05	2.43	0.016
	COVID-19 Stress x CFS	−0.06	0.02	−3.37	**0.001**
Fruits	COVID-19 Stress	0.15	0.57	0.26	0.795
	EOE	−0.82	0.51	−1.60	0.111
	CF	−0.06	0.06	−1.06	0.291
Vegetables	COVID-19 Stress	0.89	0.63	1.41	0.160
	EOE	−1.56	0.57	−2.74	**0.006**
	CFS	0.10	0.06	1.58	0.114

^1^ EOE = emotional overeating; ^2^ CFS = cognitive flexibility; ^3^
*p*-value < 0.05 are in bold.

**Table 5 nutrients-13-00901-t005:** Best-fit multinomial logistic regression models of COVID-19-related stress, emotional overeating, and cognitive flexibility associated with food intake change during COVID-19 (*n =* 428).

		Decreased Intake		Increased Intake	
Outcome	Independent Variables	Risk Ratio ^1^	95% CI	Risk Ratio ^1^	95% CI
Chocolate and Candies	COVID Stress	1.35	[0.961–1.89]	**1.50**	[1.08–2.09]
	Cognitive Flexibility Score	1.00	[0.967–1.03]	0.99	[0.961–1.03]
	Emotional Overeating	1.20	[0.879–1.64]	**1.64**	[1.22–2.2]
Cookies, Cakes, Pies, and Brownies	COVID Stress	**1.48**	[1.06–2.07]	**1.62**	[1.17–2.26]
	Cognitive Flexibility Score	1.01	[0.976–1.04]	1.03	[0.993–1.06]
	Emotional Overeating	1.12	[0.827–1.52]	**1.51**	[1.13–2.03]
Donuts, Danishes, and Muffins	COVID Stress	1.17	[0.858–1.6]	1.39	[0.961–2.01]
	Cognitive Flexibility Score	1.00	[0.965–1.03]	1.02	[0.981–1.06]
	Emotional Overeating	1.14	[0.861–1.51]	**1.56**	[1.13–2.15]
Ice Cream and Frozen Desserts	COVID Stress	**1.42**	[1.01–1.99]	**1.39**	[1–1.92]
	Cognitive Flexibility Score	0.99	[0.96–1.03]	1.01	[0.978–1.04]
	Emotional Overeating	1.31	[0.96–1.8]	**1.73**	[1.29–2.32]
Regular Chips	COVID Stress	**1.45**	[1.01–2.08]	1.24	[0.91–1.69]
	Cognitive Flexibility Score	0.98	[0.946–1.02]	1.00	[0.972–1.04]
	Emotional Overeating	1.11	[0.794–1.54]	**1.51**	[1.14–1.99]
Low-Fat Chips	COVID Stress	1.30	[0.931–1.82]	**1.71**	[1.1–2.65]
	Cognitive Flexibility Score	0.98	[0.944–1.01]	0.97	[0.928–1.01]
	Emotional Overeating	1.11	[0.827–1.5]	1.42	[0.972–2.09]
Savory Snacks	COVID Stress	**1.53**	[1.07–2.19]	1.29	[0.915–1.81]
	Cognitive Flexibility Score	0.97	[0.939–1.01]	1.01	[0.972–1.04]
	Emotional Overeating	1.15	[0.832–1.58]	**1.66**	[1.24–2.24]
Fast Food	COVID Stress	1.24	[0.901–1.69]	1.03	[0.715–1.49]
	Cognitive Flexibility Score	0.99	[0.964–1.03]	1.00	[0.965–1.04]
	Emotional Overeating	1.14	[0.855–1.53]	**1.74**	[1.25–2.43]
Fruit	COVID Stress	**1.83**	[1.29–2.58]	1.25	[0.899–1.73]
	Cognitive Flexibility Score	1.00	[0.963–1.03]	1.00	[0.968–1.03]
	Emotional Overeating	1.19	[0.88–1.6]	1.13	[0.843–1.51]
Vegetables	COVID Stress	**1.48**	[1.04–2.08]	1.34	[0.954–1.88]
	Cognitive Flexibility Score	0.98	[0.951–1.02]	1.00	[0.969–1.04]
	Emotional Overeating	1.29	[0.952–1.75]	1.31	[0.973–1.76]

Reference group = no change; ^1^ bold = *p*-values < 0.05.

## Data Availability

The data presented in this study are available on request from the corresponding author.

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
