# Peer review of "COVID-19 Stress and Food Intake: Protective and Risk Factors for Stress-Related Palatable Food Intake in U.S. Adults"

_nutrients, 2021, doi:10.3390/nu13030901_

Round 1
Reviewer 1 Report
I have a few suggestions that may help to improve the manuscript:
Methods:
1. The authors should indicate the number of approval given by Reviewer Board
2. A socioeconomic disadvantage index – could authors provide the reference for this?
3. I encourage authors to analyse associations of the Change in Food intake, as dependent variable with stress, EOE and CF as variables explaining its variation. In my opinion, the change in food intake may be better measure of pandemic stress effects on nutritional habits. EOE and CF may be good prognostic characteristics for decrease and increase in this measure in face of COVID-19.
Discussion
4. Authors should give some attention to insignificance of correlation between BMI and other variables, but also it would be great to see more comprehensive analyses of the association of BMI with food intake, psychological characteristics and stress.
5. “Notably, a number of studies show that stress impairs cognitive flexibility [50–338 53], suggesting cognitive flexibility and stress are negatively correlated, as seen in our sample.” – but in the table with correlation analysis results there are no significant correlations.
Reviewer 2 Report
I thank the editor for giving me the opportunity to review this work. The interest in the impact of the covid-19 pandemic on lifestyles and habits, including eating habits, is relevant. The authors of this study confirm both the role of the pandemic in increasing the level of stress in the population and how this situation is associated with maladaptive eating habits, characterized by an increased consumption of snacks and sweets.
The study appropriately links to an emerging literature investigating overeating, emotional eating, and poor food choices in association with the stressful COVID-19 experience. Moreover, it also introduced the variable of cognitive flexibility, which might explain some behavioral tendence of the individuals.
I did not find particular criticisms in the study. The only concern is the absence of some information about the sample. Is there any participant who was infected with covid-19 and experienced symptoms? this is because the infection could have affected in some way the consumption of food, reducing or increasing it. Please possibly discuss this. Also, about discussions I would also cite that portion of the literature that found improved eating habits during lockdown, to reinforce the possible role of cognitive flexibility.
